# ZipAR: Parallel Autoregressive Image Generation through Spatial Locality

**Yefei He** [1 2]  **Feng Chen** [3]  **Yuanyu He** [1]  **Shaoxuan He** [1]  **Hong Zhou** [1]  **Kaipeng Zhang** [2]  **Bohan Zhuang** [1]

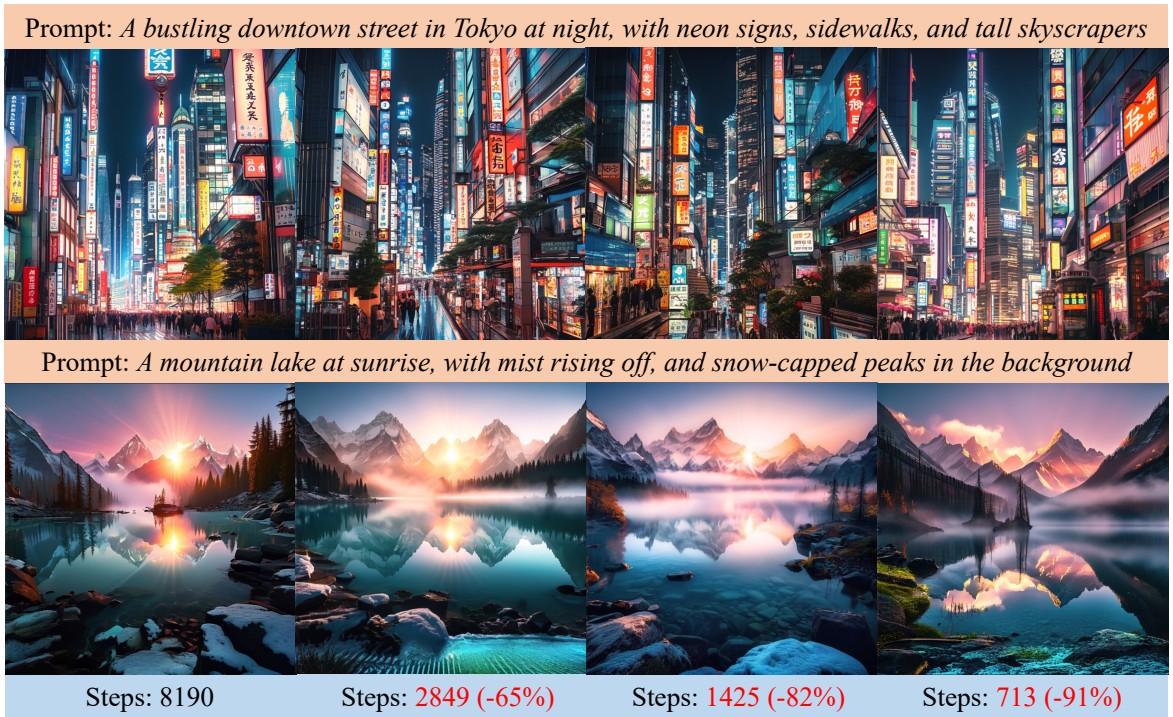

Prompt: *A bustling downtown street in Tokyo at night, with neon signs, sidewalks, and tall skyscrapers*

Prompt: *A mountain lake at sunrise, with mist rising off, and snow-capped peaks in the background*

| Steps: 8190 | Steps: 2849 (-65%) | Steps: 1425 (-82%) | Steps: 713 (-91%) |

Figure 1: **Up to 91% forward step reduction with ZipAR**. Samples are generated by Emu3-Gen model with next-token prediction paradigm (the first column) and ZipAR (the right three columns).

## Abstract

In this paper, we propose **ZipAR**, a training-free, plug-and-play parallel decoding framework for accelerating autoregressive (AR) visual generation. The motivation stems from the observation that images exhibit local structures, and spatially distant regions tend to have minimal interdependence. Given a partially decoded set of visual tokens, in addition to the original next-token prediction scheme in the row dimension, the tokens corresponding to spatially adjacent regions in the column dimension can be decoded in parallel. To ensure alignment with the contextual requirements of each token, we employ an adaptive local window assignment scheme with rejection sampling analogous to speculative decoding. By decoding multiple tokens in a single forward pass, the number of forward passes required to generate an image is significantly reduced, resulting in a substantial improvement in generation efficiency. Experiments demonstrate that ZipAR can reduce the number of model forward passes by up to $91\%$ on the Emu3-Gen model without requiring any additional retraining.

[1]Zhejiang University, China [2]Shanghai AI Laboratory, China [3]The University of Adelaide, Australia. Correspondence to: Hong Zhou <zhouhong_zju@zju.edu.cn>, Kaipeng Zhang <kp_zhang@foxmail.com>.

*Proceedings of the $42^{nd}$ International Conference on Machine Learning*, Vancouver, Canada. PMLR 267, 2025. Copyright 2025 by the author(s).

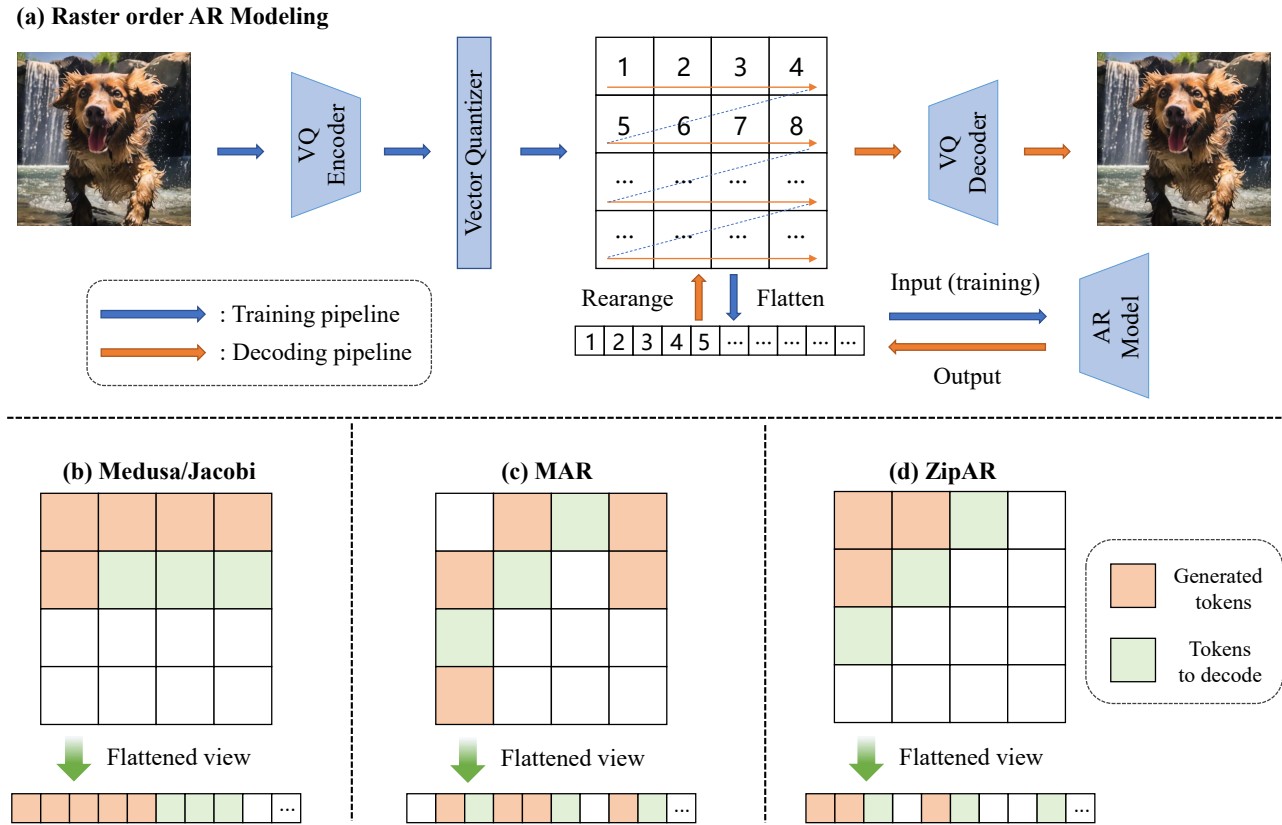

Figure 2: (a) An overview of the training and decoding pipeline for autoregressive (AR) visual generation models. For models trained with a next-token prediction objective, each forward pass generates a single visual token. (b) Medusa (Cai et al., 2024) and Jacobi (Santilli et al., 2023) decoding predict multiple adjacent tokens in sequence order. (c) MAR (Li et al., 2024) predicts multiple tokens in a random order. (d) The proposed ZipAR predicts multiple spatially adjacent tokens.

## 1. Introduction

Recent advancements in large language models (LLMs) with the "next-token prediction" paradigm (Achiam et al., 2023; Vavekanand & Sam, 2024; Team et al., 2023) have demonstrated remarkable capabilities in addressing text-related tasks. Building on these successes, many studies (Liu et al., 2024a; Wang et al., 2024b; Team, 2024; Ge et al., 2024; Wu et al., 2024a) have extended this paradigm to the generation of visual content, leading to the development of autoregressive (AR) visual generation models. These models not only produce high-fidelity images and videos that rival or even exceed the performance of state-of-the-art diffusion models but also facilitate unified multimodal understanding and generation (Wang et al., 2024a; Chen et al., 2025; Wu et al., 2024a;b). However, their slow generation speed remains a significant barrier to widespread adoption. To generate high-resolution images or videos, these models must sequentially produce thousands of visual tokens, requiring numerous forward passes and resulting in high latency.

To reduce the number of forward passes required for generating lengthy responses, several studies (Cai et al., 2024; Santilli et al., 2023; Chen et al., 2023) have proposed the "next-set prediction" paradigm for LLMs, as depicted in Figure 2(b). These approaches involves introducing multiple decoding heads (Cai et al., 2024) or small draft models (Chen et al., 2023), which generate several candidate tokens that are later evaluated by the original model. However, these methods incur additional costs, as they require extra draft models or the training of new decoding heads. Another approaches use the jacobi decoding methods (Santilli et al., 2023; Fu et al., 2024; Teng et al., 2024), iteratively updates sequences of tokens until convergence. However, in practice, the acceleration achieved by these methods is marginal, as LLMs often fail to generate correct tokens when errors exist in preceding ones. Furthermore, none of these approaches exploit the unique characteristics of visual content, and a parallel decoding framework specifically tailored for AR visual generation has yet to be developed.

In this paper, we introduce ZipAR, a parallel decoding framework designed to accelerate AR visual generation.

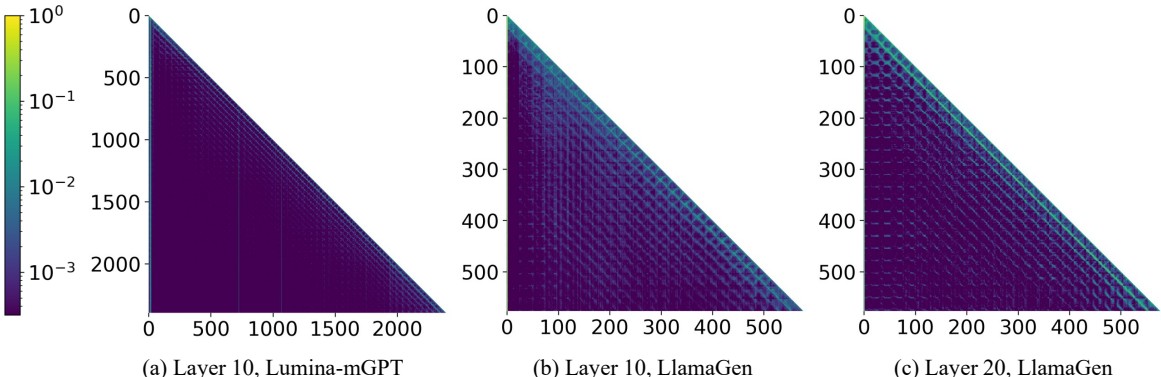

Figure 3: The attention scores of visual tokens in the Lumina-mGPT-7B (Liu et al., 2024a) and LlamaGen-XL (Sun et al., 2024) models. **Slash lines indicate that significant attention scores are allocated to tokens at fixed intervals, corresponding to tokens in the same column of previous rows.** The full attention scores are presented by storing the attention scores of each visual token during decoding and concatenating them.

As depicted in Figure 2(a), common AR visual generation models produce visual tokens in a raster order, where the first token in a row cannot be generated until the last token in the preceding row is decoded despite their spatial separation. However, visual content inherently exhibits strong locality, which is a widely utilized inductive bias for visual tasks (Liu et al., 2021; Zhang et al., 2022; LeCun et al., 1989; Krizhevsky et al., 2012; Zeiler & Fergus, 2014). Specifically, there are significant spatial correlations between spatially adjacent tokens (e.g., token 5 and token 1 in Figure 2(a)) compared to tokens that are adjacent only in the generation order (e.g., token 5 and token 4), which makes the raster-order sequential dependency suboptimal. Empirical evidence, as shown in Figure 3, further supports this observation, with significant attention allocated to tokens in the same column of the previous row. This motivates us to propose **decoding tokens from the next row without waiting for the full decoding of the current row**, enabling the parallel decoding of multiple tokens in a single forward pass. Specifically, a predefined window size determines whether two tokens are spatially adjacent. Tokens outside this window in adjacent rows are considered irrelevant. Consequently, once the number of generated tokens in a row exceeds the window size, decoding of the next row begins in parallel with the current row. With an appropriately chosen window size, multiple rows can be decoded simultaneously. Unlike Medusa (Cai et al., 2024), which employs auxiliary heads, all tokens generated in parallel by ZipAR are produced using the original model head. Moreover, to address the limitation that manually tuned window size may not optimally adapt to varying attention distributions across tokens, we introduce an adaptive window size assignment scheme. This scheme dynamically adjusts the local window size during generation, ensuring that each token is generated with a window size tailored to its contextual requirements. As a result, ZipAR can be seamlessly implemented in a training-free, plug-and-play manner for autoregressive visual generation models, without introducing additional overhead. Experiments across multiple autoregressive visual generation models demonstrate the effectiveness and robustness of ZipAR, achieving forward steps reductions of 91%, 75%, and 81% on Emu3-Gen, Lumina-mGPT-7B, and LlamaGen-XL models, respectively, with minimal degradation in image quality.

In summary, our contributions are as follows:

- We propose a spatially-aware parallel decoding strategy that enables inter-row token generation by leveraging the inherent spatial locality of visual content. Once the number of generated tokens in a row exceeds a window size, decoding of the next row begins in parallel.

- We propose an adaptive window size assignment scheme that dynamically adjusts the local window size for each token during generation, optimizing decoding efficiency while ensuring the contextual information essential for producing high-quality tokens.

- By integrating these techniques, we present ZipAR, a training-free, plug-and-play framework that achieves significant acceleration in autoregressive visual generation. Extensive experiments demonstrate its effectiveness and robustness across multiple AR visual generation models.

## 2. Related Work

### 2.1. Autoregressive Visual Generation

The success of Transformer models in text-based tasks has inspired studies (Van Den Oord et al., 2017; Esser et al.,

2021; Yu et al., 2023) to apply autoregressive modeling to visual content generation. These methods can be classified into two main categories: GPT-style approaches that utilize the next-token prediction paradigm (Esser et al., 2021; Wang et al., 2024b; Liu et al., 2024a; Sun et al., 2024) and BERT-style approaches that employ masked prediction models (Chang et al., 2022; 2023; Li et al., 2024; Yu et al., 2023). More recently, VAR (Tian et al., 2024) modified the traditional next-token prediction paradigm to next-scale prediction, resulting in faster sampling speeds. Models trained using next-token prediction can leverage the infrastructure and training techniques of large language models (LLMs) and pave the way towards unified multi-modal understanding and generation. However, they are generally less efficient during sampling compared to models that predict multiple tokens in a single forward pass. In this paper, we focus on accelerating visual generation models trained with the next-token prediction objective, hereafter referred to as autoregressive visual generation models unless otherwise specified.

### 2.2. Efficient Decoding of LLMs.

Efforts to reduce the number of forward passes required for LLMs to generate lengthy responses can be broadly categorized into two main approaches. The first approach involves sampling multiple candidate tokens before verifying them with the base LLM. Speculative decoding (Chen et al., 2023; Liu et al., 2024b; Spector & Re, 2023; Gui et al., 2024) utilizes a small draft LLM to generate candidate tokens, which are then verified in parallel by the base LLM. While this approach can potentially generate multiple tokens in a single evaluation, deploying multiple models introduces significant memory overhead and engineering challenges. Medusa (Cai et al., 2024) addresses this by employing multiple decoding heads for the base LLM, enabling self-speculation. However, due to the large vocabulary size of LLMs, the parameters in each decoding head can be substantial. The second approach, Jacobi decoding (Santilli et al., 2023; Teng et al., 2024), involves randomly guessing the next n tokens in a sequence, which are iteratively updated by the LLMs. Over time, the n-token sequence converges to the same output as that generated by the next-token prediction paradigm. However, in practice, vanilla Jacobi decoding offers only marginal speedup over autoregressive decoding. This limited improvement is largely due to the causal attention mechanism, which rarely produces a correct token when preceding tokens are incorrect. Lookahead (Fu et al., 2024) decoding enhances efficiency by leveraging n-grams generated from previous Jacobi iterations, which are verified in parallel during the decoding process. CLLMs (Kou et al., 2024) further improves the efficiency of Jacobi decoding by fine-tuning the model with a consistency loss, requiring it to map arbitrary

points on the Jacobi trajectory to a fixed point. However, none of these approaches are designed for autoregressive visual generation or incorporate visual inductive biases. In contrast, the proposed ZipAR takes advantage of the spatial locality inherent in visual content, offering significant acceleration without the need for retraining. Moreover, ZipAR is orthogonal to the aforementioned methods, and can be combined with them to achieve even greater acceleration.

## 3. Method

### 3.1. Preliminaries

Autoregressive (AR) visual generation models with the next-token prediction paradigm have shown exceptional versatility across various vision-language tasks, including generating high-quality images and videos. As shown in Figure 2(a), pre-trained VQ-VAE models (Van Den Oord et al., 2017; Esser et al., 2021) are commonly employed to convert images or videos into visual tokens. The process begins with a visual encoder that extracts feature maps at a reduced spatial resolution. These feature maps are then subjected to vector quantization to produce discrete latent representations, known as visual tokens. These tokens are arranged in a one-dimensional sequence to serve as input for AR models. Although various methods exist to flatten these tokens, the row-major order (raster order) is empirically validated to offer the best performance (Esser et al., 2021), making it the prevalent method for visual generation. During the image generation phase, AR models generate visual tokens sequentially in this raster order. Finally, the complete sequence of visual tokens is rearranged into a two-dimensional structure and processed through a visual decoder to reconstruct the images.

### 3.2. Inference with ZipAR

As analyzed in Section 3.1, AR visual generation models with a raster order generate visual tokens row by row, completing each row sequentially from left to right before proceeding to the next. However, images inherently exhibit strong spatial locality. Intuitively, in a high-resolution image, the starting pixel of a row is more closely related to the starting pixel of the preceding row than to the ending pixel of the preceding row due to their spatial proximity. Empirical evidence, as shown in Figure 3, also indicates that significant attention scores are allocated to tokens within the same column of the previous row. Building on these observations, we propose **ZipAR**, a simple yet effective parallel decoding framework for autoregressive visual generation models. Unlike conventional parallel decoding methods that predict multiple consecutive tokens in a single forward pass, our approach decodes tokens from different rows in parallel. The key idea is that *it is unnecessary to wait for an entire row to be generated before initiating the decoding of the*

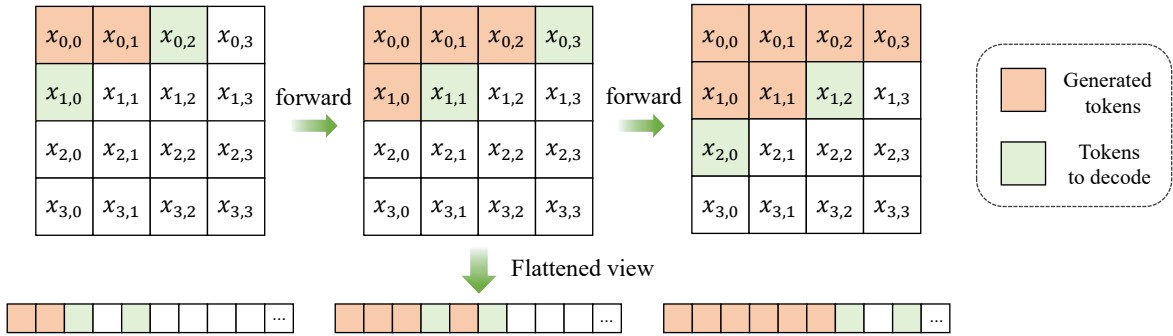

Figure 4: A toy example of the ZipAR framework. The window size is set to 2 in this toy example.

*next row*, as spatially distant tokens contribute minimally to attention scores.

To formalize this, we define a local window size $s$. Given the tokens $x_{i,j}$ located in row $i$ and column $j$, we assume that tokens beyond $x_{i-1,j+s}$ in the previous row have a negligible impact on the generation of $x_{i,j}$ based on the spatial locality of visual tokens. Consequently, the criterion for initiating the generation of token $x_{i,j}$ can be formulated as:

$$C(i,j) = \begin{cases} 1, & \text{if } \{x_{i-1,k} \mid j \le k < j+s\} \subseteq \mathbb{D} \\ 0, & \text{otherwise} \end{cases} \quad (1)$$

Here, $\mathbb{D}$ denotes the set of decoded tokens, and $C(i,j) = 1$ indicates that token $x_{i,j}$ is ready to be generated. Once the first token in a row is generated, subsequent tokens in the row can be generated sequentially, along with the unfinished portion of the preceding row, following a next-token prediction paradigm. An illustration of the ZipAR framework with a window size of 2 is shown in Figure 4.

However, to initiate the decoding of the first token $x_{i,0}$ in row $i$, the last token of the row $i-1$ is required as input to the autoregressive model, despite it has not yet been generated in the ZipAR framework. To address this, we propose several solutions tailored to different types of AR visual generation models. Some methods (Liu et al., 2024a; Wang et al., 2024b) support generating images with dynamic resolutions, typically by appending extra end-of-row tokens at the end of each row. With these special tokens placed at fixed positions, we can insert the end-of-row tokens in advance when initiating the generation of the next row. Since the values of these tokens are predetermined, there is no need to update them subsequently. Conversely, for models that lack end-of-row tokens (Sun et al., 2024), we temporarily assign values to the last token in row $i-1$ to decode token $x_{i,0}$. This value can be derived from the most spatially adjacent token that have been decoded.

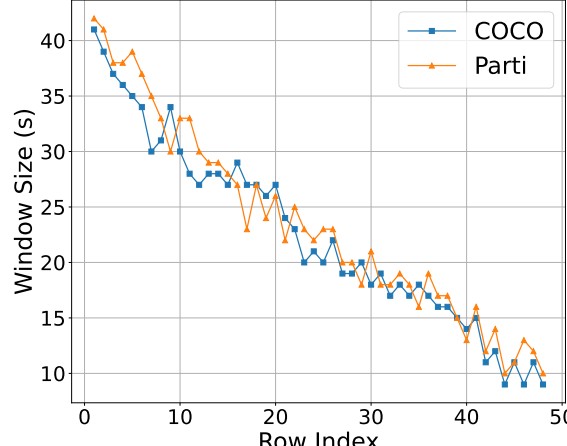

Figure 5: The local window size required to retain 95% of attention scores across different rows and input prompt. Data is collected from the first token of each row in Lumina-mGPT-7B model with input prompt from COCO (Lin et al., 2014) and Parti (Yu et al., 2022) dataset.

### 3.3. Adaptive Window Size Assignment

While ZipAR with a predefined local window size demonstrates improved efficiency, the window size remains a hyperparameter that requires manual tuning to balance image fidelity and generation efficiency. Moreover, using a fixed window size for all token positions is suboptimal, as the attention distributions vary significantly across tokens. As illustrated in Figure 5, the local window size needed to retain 95% of attention scores differs across token positions and input prompts. Consequently, maintaining a fixed window size throughout the image generation process can lead to suboptimal results, potentially compromising image fidelity.

To address this, we propose an adaptive window size assignment scheme that dynamically adjusts the local window size during the generation process. Given a minimum window size $s_{min}$, after generating token $x_{i,s_{min}-1}$ in row $i$, we attempt to generate the first token in row $i+1$. Unlike the

fixed window size approach, we do not immediately accept this newly generated token, as the current local window size may provide insufficient information. Instead, in the subsequent step, with the addition of a new token from the previous row, we regenerate the token using a slightly larger window size $s_{min} + 1$ and apply an acceptance criterion to evaluate its validity based on the predictions from both steps. If the criterion is satisfied, subsequent tokens in row $i + 1$ can be generated sequentially, following a next-token prediction paradigm. Otherwise, the current window size is deemed inadequate, and we iteratively expand it until the criterion is met or the previous row is fully generated.

Specifically, we adopt a rejection sampling scheme analogous to speculative decoding (Leviathan et al., 2023; Chen et al., 2023). For consecutive window sizes $k + 1$ and $k$ in row $i$, we compute the ratio between their predictions $p(x|x_{0,0}, ..., x_{i,k})$ and $p(x|x_{0,0}, ..., x_{i,k-1})$, which quantifies how well the token sampled under the smaller window size. Formally, the criterion for initiating the generation of token $x_{i+1,0}$ with window size $k$ can be formulated as:

$$\tilde{C}(i + 1, 0) = \begin{cases} 1, & \text{if } r < \min\left(1, \frac{p(x|x_{0,0}, ..., x_{i,k})}{p(x|x_{0,0}, ..., x_{i,k-1})}\right), \\ 0, & \text{otherwise} \end{cases}$$
(2)

Here, we sample $r \sim U[0, 1]$ from a uniform distribution. $\tilde{C}(i + 1, 0) = 1$ indicates that token $x_{i+1,0}$ is ready to be generated. If the criterion is not met, we resample $x_{i+1,0}$ from the following distribution:

$$x_{i+1,0} \sim \frac{\max(0, p(x|x_{0,0}, ..., x_{i,k}) - p(x|x_{0,0}, ..., x_{i,k-1}))}{\sum_x \max(0, p(x|x_{0,0}, ..., x_{i,k}) - p(x|x_{0,0}, ..., x_{i,k-1}))}$$
(3)

The resampled token is subsequently verified in the next step.

# 4. Experiments

## 4.1. Implementation Details

To assess the effectiveness of our proposed method, we integrate it with three state-of-the-arts autoregressive visual generation models: LlamaGen (Sun et al., 2024), Lumina-mGPT (Liu et al., 2024a) and Emu3-Gen (Wang et al., 2024b). All experiments are conducted with Nvidia A100 GPUs and Pytorch framework. For class-conditional image generation with LlamaGen on ImageNet, we report the widely adopted Frechet Inception Distance (FID) to evaluate the performance. We sample 50000 images and evaluate them with ADM's TensorFlow evaluation suite (Dhariwal & Nichol, 2021).

## 4.2. Main Results

### 4.2.1. CLASS-CONDITIONAL IMAGE GENERATION

In this subsection, we quantitatively evaluate the performance of class-conditional image generation on the ImageNet $256 \times 256$ benchmark using the LlamaGen model, as summarized in Table 1. The model processes a $24 \times 24$ feature map and requires 576 forward passes to generate an image under the next-token prediction (NTP) paradigm. For the LlamaGen-L model, integrating ZipAR with a minimal window size of 16 reduces the number of forward passes by 26.7% without increasing the FID score. For the LlamaGen-XL model, ZipAR-12 achieves a lower FID (3.67 vs. 3.87) while requiring fewer steps than the previous parallel decoding algorithm, SJD (Teng et al., 2024) (331 steps vs. 335 steps). This highlights the efficiency of ZipAR in decoding spatially adjacent tokens in parallel.

### 4.2.2. TEXT-GUIDED IMAGE GENERATION

In this subsection, we expand our evaluation by assessing ZipAR's performance using multiple metrics, including VQAScore (Lin et al., 2024), Human Preference Score v2 (HPSv2) (Wu et al., 2023), ImageReward (Xu et al., 2023), and Aesthetic Score, across three models: LlamaGen-XL-512, Lumina-mGPT-768, and Lumina-mGPT-1024, as presented in Table 2. For the LlamaGen-XL model, ZipAR-15 reduces the number of generation steps by 45.1% without any decline in the VQAScore, Image Reward and Aesthetic Score. Similarly, for the Lumina-mGPT-768 model, ZipAR-20 achieves a 54.8% reduction in generation steps while improving VQAScore, HPSv2, and Aesthetic Score. When evaluating the CLIP Score over the LlamaGen-XL model, compared to the previous parallel decoding algorithm, SJD (Teng et al., 2024), ZipAR-7 significantly improves efficiency (324 steps vs. 635 steps) while achieving a higher CLIP score (0.285 vs. 0.283). Moreover, we observe that the acceleration ratio for both text-to-image models is higher than that for the class-conditional LlamaGen-L model. This is primarily attributed to the larger spatial resolution of the feature maps and the generated images. These results suggest that ZipAR provides greater efficiency gains when generating higher-resolution images.

## 4.3. Ablation Study

### 4.3.1. EFFECT OF ADAPTIVE WINDOW SIZE ASSIGNMENT

In this subsection, we evaluate the effectiveness of the proposed adaptive window size assignment scheme. Specifically, we compare the performance of ZipAR with fixed and adaptive window sizes over class-conditional LlamaGen-L model, respectively. As shown in Figure 6, under similar generation steps, ZipAR with adaptive window size

Table 1: Quantitative evaluation on ImageNet $256 \times 256$ benchmark. The generated images are $384 \times 384$ and resized to $256 \times 256$ for evaluation. Here, "NTP" denotes the next-token prediction paradigm. "ZipAR-$n$" denotes the ZipAR paradigm with a minimal window size of $n$. "Step" is the number of model forward passes required to generate an image. The latency is measured with a batch size of 1.

| Model | Method | Step | Latency (s) | FID↓ |
|---|---|---|---|---|
| | NTP | 576 | 15.20 | 3.16 |
| | SJD (Teng et al., 2024) | 367 (-36.3%) | 10.83 (-28.8%) | 3.85 |
| LlamaGen-L (cfg=2.0) | ZipAR-16 | 422 (-26.7%) | 11.31 (-25.6%) | **3.14** |
| | ZipAR-14 | 378 (-34.4%) | 10.16 (-33.2%) | 3.44 |
| | ZipAR-12 | **338 (-41.3%)** | **9.31 (-38.8%)** | 3.96 |
| | NTP | 576 | 22.65 | **2.83** |
| | SJD (Teng et al., 2024) | 335 (-41.8%) | 13.17 (-41.8%) | 3.87 |
| LlamaGen-XL (cfg=2.0) | ZipAR-16 | 423 (-26.6%) | 16.46 (-27.3%) | 2.87 |
| | ZipAR-14 | 378 (-34.4%) | 14.89 (-34.3%) | 3.16 |
| | ZipAR-12 | **331 (-41.8%)** | **13.17 (-41.8%)** | 3.67 |

Table 2: Quantitative results on diverse automatic evaluation approaches. Here, "NTP" denotes the next-token prediction paradigm. "ZipAR-$n$" denotes the ZipAR paradigm with a minimal window size of $n$. "Step" is the number of model forward passes required to generate an image.

| Model | Method | Steps | VQAScore↑ | HPSv2↑ | Image Reward↑ | Aesthetic Score↑ |
|---|---|---|---|---|---|---|
| | NTP | 1024 | 0.6439 | **0.2647** | -0.0818 | 5.38 |
| | ZipAR-15 | 562 | 0.6534 | 0.2637 | **-0.0690** | **5.39** |
| LlamaGen-XL-512 | ZipAR-11 | 451 | **0.6581** | 0.2630 | -0.0982 | 5.37 |
| | ZipAR-7 | 324 | 0.6410 | 0.2625 | -0.1683 | 5.33 |
| | ZipAR-3 | 185 | 0.6343 | 0.2599 | -0.3121 | 5.32 |
| | NTP | 2352 | 0.6579 | 0.2743 | **0.4164** | 6.10 |
| | ZipAR-20 | 1063 | **0.6595** | **0.2747** | 0.3971 | **6.13** |
| Lumina-mGPT-768 | ZipAR-17 | 915 | 0.6433 | 0.2732 | 0.3049 | 6.12 |
| | ZipAR-14 | 740 | 0.6589 | 0.2739 | 0.3646 | 6.10 |
| | ZipAR-11 | 588 | 0.6490 | 0.2730 | 0.2861 | 6.10 |
| | NTP | 4160 | 0.6718 | **0.2762** | **0.4232** | **5.97** |
| | ZipAR-20 | 1331 | 0.6705 | 0.2761 | 0.3913 | 5.95 |
| Lumina-mGPT-1024 | ZipAR-17 | 1150 | **0.6797** | 0.2761 | 0.4018 | 5.94 |
| | ZipAR-14 | 964 | 0.6732 | 0.2747 | 0.3298 | 5.94 |
| | ZipAR-11 | 772 | 0.6723 | 0.2746 | 0.3222 | 5.95 |

consistently achieves a lower FID than its fixed-window counterpart, which suggests that dynamically adjusting the window size based on token position and context enhances the fidelity of generated images.

### 4.3.2. IMPACT ON OPTIMAL SAMPLING HYPERPARAMETERS

As presented in Tables 4-5, we performed a grid search to determine the optimal token-sampling hyperparameters, namely, sampling temperature and classifier-free guidance scale, for ZipAR. The results are shown below. These results indicate that ZipAR sampling does not alter the optimal sampling temperature and classifier-free guidance scale.

### 4.4. Qualitative Visualizations

In this subsection, we present non-cherry-picked visualizations of images generated using the next-token prediction

(NTP) paradigm and the proposed ZipAR framework over Emu3-Gen (Wang et al., 2024b) and Lumina-mGPT-7B (Liu et al., 2024a), as shown in Figures 1 and 7. Notably, ZipAR can reduce the number of model forward steps by up to 91% for Emu3-Gen, while still producing high-fidelity images rich in semantic information.

## 5. Conclusion

In this paper, we have proposed ZipAR, a new parallel decoding framework designed to accelerate autoregressive visual generation. ZipAR leverages the spatial locality inherent in visual content and predicts multiple spatially adjacent visual tokens in a single model forward pass, thereby significantly enhancing generation efficiency compared to the traditional next-token-prediction paradigm. An adaptive local window assignment scheme with rejection sampling is employed, ensuring that each token is generated with

Table 3: Quantitative evaluation on MS-COCO dataset. Here, "NTP" denotes the next-token prediction paradigm. "ZipAR-$n$" denotes the ZipAR paradigm with a minimal window size of $n$. "Step" is the number of model forward passes required to generate an image. The latency is measured with a batch size of 1.

| Model | Method | Step | Latency (s) | CLIP Score↑ |
|---|---|---|---|---|
| | NTP | 1024 | 33.17 | 0.287 |
| | SJD (Teng et al., 2024) | 635 (-38.0%) | 24.80 (-25.2%) | 0.283 |
| LlamaGen-XL-512 | ZipAR-15 | 562 (-45.1%) | 18.98 (-42.7%) | **0.287** |
| | ZipAR-11 | 451 (-55.9%) | 14.65 (-55.8%) | 0.286 |
| | ZipAR-7 | 324 (-68.4%) | 10.24 (-69.1%) | 0.285 |
| | ZipAR-3 | **185 (-81.9%)** | **5.86 (-82.3%)** | 0.281 |
| | NTP | 2352 | 91.70 | 0.313 |
| | SJD (Teng et al., 2024) | 1054 (-55.2%) | 60.27 (-34.2%) | 0.313 |
| Luming-mGPT-7B-768 | ZipAR-20 | 1063 (-54.8%) | 63.28 (-31.0%) | **0.314** |
| | ZipAR-17 | 915 (-61.0%) | 58.54 (-36.2%) | 0.314 |
| | ZipAR-14 | 740 (-68.5%) | 53.41 (-41.8%) | 0.313 |
| | ZipAR-11 | **588 (-75.0%)** | **50.32 (-45.1%)** | 0.312 |

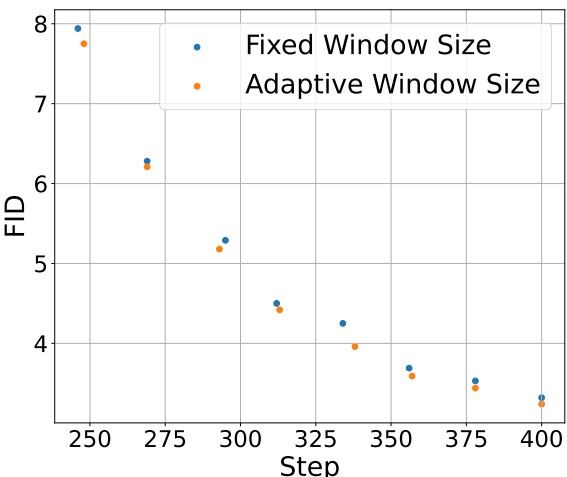

Figure 6: Performance comparisons of ZipAR over class-conditional LlamaGen-L model with fixed window size and adaptive window size. Under similar step budget, ZipAR with adaptive window size always achieves lower FID.

Table 4: The performance of LlamaGen and ZipAR under different classifier-free guidance. Here, "*" denotes the results obtained from LlamaGen's paper.

| Model | Classifier-free Guidance | FID↓ |
|---|---|---|
| | 1.5 | 4.74 |
| LlamaGen-L* | 1.75 | 3.15 |
| | 2.0 | **3.07** |
| | 2.25 | 3.62 |
| | 1.5 | 6.18 |
| ZipAR-16 | 1.75 | 3.72 |
| | 2.0 | **3.14** |
| | 2.25 | 3.44 |

Table 5: The performance of LlamaGen and ZipAR under different sampling temperatures. Here, "*" denotes the results obtained from LlamaGen's paper.

| Model | Temperature | FID↓ |
|---|---|---|
| | 0.96 | 3.53 |
| LlamaGen-L | 0.98 | 3.24 |
| | 1.0* | **3.07** |
| | 1.02 | 3.14 |
| | 0.96 | 3.35 |
| ZipAR-16 | 0.98 | 3.25 |
| | 1.0 | **3.14** |
| | 1.02 | 3.34 |

sufficient contextual information. Extensive experiments demonstrate that ZipAR can reduce the number of model forward steps by up to $91\%$ on the Emu3-Gen model with minimal impact on image quality.

In the future, we anticipate that integrating ZipAR with other methods that employ the next-set-prediction paradigm, such as Medusa (Cai et al., 2024) and Jacobi decoding (Santilli et al., 2023), will further enhance acceleration ratios.

## Impact Statement

The proposed ZipAR framework stands out for its high efficiency, which carry significant implications in reducing the carbon emissions attributed to the widespread deployment of deep generative models. However, similar to other deep generative models, ZipAR has the potential to be utilized for producing counterfeit images and videos for malicious purposes.

## Acknowledgements

This work was supported by the National Key Research and Development Program of China (2022YFC3602601) and the National Key Research and Development Program of China (2022ZD0160102).

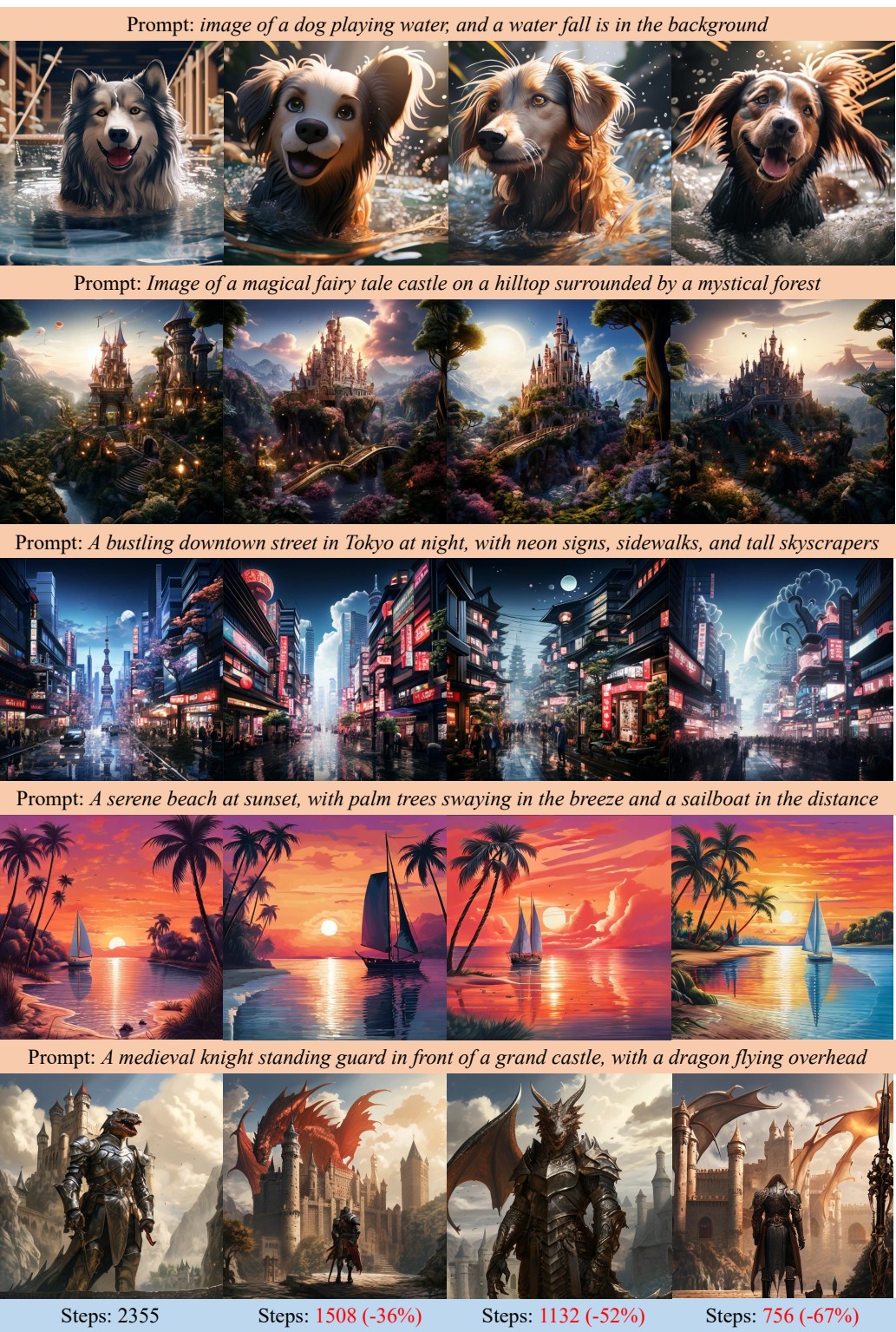

Figure 7: Samples generated by the Lumina-mGPT-7B-768 model with next-token prediction paradigm (the first column) and ZipAR under different configurations (the right three columns). The classifier-free guidance is set to 3.

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
