# OpenReview forum: "ZipAR: Parallel Autoregressive Image Generation through Spatial Locality"
_ICML.cc/2025/Conference — ICML 2025 poster_

### Official Review · Reviewer_b59B · 2025-03-07

**Overall Recommendation:** 3

**Summary:**

The paper proposes ZipAR, a training-free plug-and-play decoding method for accelerating auto-regressive visual generation models. It decodes spatially adjacent tokens in the column dimension in parallel. It emplys an adaptive local window assignment scheme with reject sampling strategy. Experimental results demonstrate that ZipAR can decrease the number of required forward passes without compromising the quality of generation.

## Update after rebuttal

The rebuttal satisfactorily addresses my concerns regarding the evaluation and theoretical justification. However, I remain concerned about the noticeable artifacts in the images. As a result, I am increasing my score to Weak Accept.

**Claims And Evidence:**

Yes.

**Essential References Not Discussed:**

NA

**Experimental Designs Or Analyses:**

I check the soundness/validity of experimental designs and analyses. Please find my concerns in Evaluation Criteria setting.

**Methods And Evaluation Criteria:**

1. I think the evaluation is not enough. Authors only use CLIP to evlauate the quality of text-to-image generation, which fails to evaluate the image appearance. I would suggest authors to consider Aesthetic Score, Human Preference Score v2, and ImageReward to fully evaluate the method.

2. I see that ZipAR with diverse window sizes (for example from 3 to 15 in LlamaGen-XL) achieves similar CLIP score. Can you also show some qualitative comparison to see if different window sizes will lead to different behaviors?

**Other Comments Or Suggestions:**

NA

**Other Strengths And Weaknesses:**

As the number of steps decreases, some artifacts become noticeable in Fig. 8. Does ZipAR enhance efficiency at the cost of visual quality? I recommend that the authors utilize additional metrics, as outlined in the Methods and Evaluation Criteria section, to further investigate this trade-off.

**Questions For Authors:**

NA

**Relation To Broader Scientific Literature:**

NA

**Theoretical Claims:**

A theoretical proof demonstrating the effectiveness of speculative decoding in identifying short local window sizes that lead to insufficient information is currently lacking.

---

> ### Author Rebuttal · Authors · 2025-03-31
>
> Thanks to the reviewer for the valuable comments.
>
> **Q1: Utilize additional metrics to fully evaluate the method.**
> To address this concern, we have expanded our evaluation by assessing ZipAR's performance using multiple metrics, including VQAScore, Human Preference Score v2, ImageReward, and Aesthetic Score, across three models: LlamaGen-XL-512, Lumina-mGPT-768, and Lumina-mGPT-1024. The results presented below demonstrate that our method significantly improves generation efficiency with little impact on output quality across various benchmarks.
>
> | Model | Method | Steps | VQAScore | HPSv2 | Image Reward | Aesthetic Score |
> | ---- | ---- | ---- | ---- | ---- | ---- | ---- |
> | LlamaGen-XL | NTP | 1024 | 0.6439 | **0.2647** | -0.0818 | 5.38 |
> | LlamaGen-XL | ZipAR-15 | 562 | 0.6534 | 0.2637 | **-0.0690** | **5.39** |
> | LlamaGen-XL | ZipAR-11 | 451 | **0.6581** | 0.2630 | -0.0982 | 5.37 |
> | LlamaGen-XL | ZipAR-7 | 324 | 0.6410 | 0.2625 | -0.1683 | 5.33 |
> | LlamaGen-XL | ZipAR-3 | 185 | 0.6343 | 0.2599 | -0.3121 | 5.32 |
> | Lumina-mGPT-768 | NTP | 2352 | 0.6579 | 0.2743 | **0.4164** | 6.10 |
> | Lumina-mGPT-768 | ZipAR-20 | 1063 | **0.6595** | **0.2747** | 0.3971 | **6.13** |
> | Lumina-mGPT-768 | ZipAR-17 | 915 | 0.6433 | 0.2732 | 0.3049 | 6.12 |
> | Lumina-mGPT-768 | ZipAR-14 | 740 | 0.6589 | 0.2739 | 0.3646 | 6.10 |
> | Lumina-mGPT-768 | ZipAR-11 | 588 | 0.6490 | 0.2730 | 0.2861 | 6.10 |
> | Lumina-mGPT-1024 | NTP | 4160 | 0.6718 | **0.2762** | **0.4232** | **5.97** |
> | Lumina-mGPT-1024 | ZipAR-20 | 1331 | 0.6705 | 0.2761 | 0.3913 | 5.95 |
> | Lumina-mGPT-1024 | ZipAR-17 | 1150 | **0.6797** | 0.2761 | 0.4018 | 5.94 |
> | Lumina-mGPT-1024 | ZipAR-14 | 964 | 0.6732 | 0.2747 | 0.3298 | 5.94 |
> | Lumina-mGPT-1024 | ZipAR-11 | 772 | 0.6723 | 0.2746 | 0.3222 | 5.95 |
>
> **Q2: Show some qualitative comparisons of different window sizes.**
> We have provided qualitative visualizations of different window sizes, as referred to Figures 1, 8 in the paper and Figures 9, 10 in the supplementary material. Since we can not update PDF version in the current phase, we will add more qualitative comparisons of different window sizes in the revised version.
>
> **Q3: The effectiveness of speculative decoding in identifying short local window sizes that lead to insufficient information.**
> Let $p_s$ and $p_{s+1}$ be the token distributions for window sizes $s$ and $s+1$. As referred to Eq. 2 in the paper, the acceptance probability for a candidate $x_s \sim p_s$ is defined as
> $$
> \alpha(x_s)= \min\left(1, \frac{p_{s+1}(x_s)}{p_s(x_s)}\right)
> $$
>
> The expectation of the acceptance can be formulated as:
> $$
> \mathbb{E}_{x \sim p_s}\left[\alpha(x_s)\right]
> $$
>
> $$
> = \sum_x p_s(x) \min(1, \frac{p_{s+1}(x)}{p_s(x)}) = \sum_x \min(p_s(x), p_{s+1}(x))
> $$
>
> **Theorem 1** (Relationship between Pairwise Minimum and Total Variation).
> Let $p$ and $q$ be two probability distributions over the same discrete support $\mathcal{X}$. The sum of their element-wise minima satisfies:
> $$
> \sum_{x \in \mathcal{X}} \min(p(x), q(x)) = 1 - \text{TV}(p, q),
> $$
> where $\text{TV}(p, q) = \frac{1}{2} \|p - q\|_1$ is the total variation distance between $p$ and $q$.
>
> **Proof**
>
> For any $x \in \mathcal{X}$, observe that:
> $$
> \max(p(x), q(x)) + \min(p(x), q(x)) = p(x) + q(x).
> $$
> Summing over all $x$ yields:
> $$
> \sum_{x} \max(p(x), q(x)) + \sum_{x} \min(p(x), q(x)) = \sum_{x} p(x) + \sum_{x} q(x) = 2.
> \quad (1)
> $$
>
> By definition of the L1-norm, we have:
> $$
> \|p - q\|_1
> $$
>
> $$
> = \sum_{x} \|p(x) - q(x)\|
> $$
>
> $$
> = \sum_{x} \max(p(x), q(x)) - \sum_{x} \min(p(x), q(x)).
> \quad (2)
> $$
>
> Let $S_{\min} = \sum_{x} \min(p(x), q(x))$ and $S_{\max} = \sum_{x} \max(p(x), q(x))$.
> Then Equation (1) can be reformulated as:
> $$
> S_{\max} = 2 - S_{\min}. \quad (3)
> $$
>
> By substituting Equation (3) into (2), we have:
> $$
> \|p - q\|_1
> $$
>
> $$
> = (2 - S_{\min}) - S_{\min}
> $$
>
> $$
> = 2 - 2S_{\min}.
> $$
> Rearranging and using $\text{TV}(p, q) = \frac{1}{2} \|p - q\|_1$:
>
> $$
> S_{\min} = 1 - \text{TV}(p, q). \quad
> $$
>
> Therefore, the expected acceptance rate is formulated as:
>
> $$
> \mathbb{E}_{x \sim p_s}\left[\alpha(x_s)\right] =
> $$
>
> $$
> \sum_x \min(p_s(x), p_{s+1}(x))=1-\text{TV}(p_{s+1},p_s) \quad (3)
> $$
>
> If $s$ is insufficient, the distributions $p_s$ and $p_{s+1}$ diverge significantly, implying $\text{TV}(p_s, p_{s+1}) \geq \Delta$ for a threshold $\Delta > 0$. By (3), the expected acceptance rate is upper bounded by:
> $$
> \mathbb{E}[\alpha] \leq 1 - \Delta.
> $$
> A low acceptance rate ($\leq 1 - \Delta$) prompts the algorithm to increase the window size to $s+1$.
>
> If $s$ is sufficient, $p_s$ and $p_{s+1}$ are statistically indistinguishable ($\text{TV}(p_s, p_{s+1}) \approx 0$). By (3), the expected acceptance rate approaches 1:
> $$
> \mathbb{E}[\alpha] \geq 1 - \epsilon \quad (\epsilon \approx 0),
> $$
> allowing immediate sampling from $p_{s+1}$ without window expansion.

---

### Official Review · Reviewer_CC9X · 2025-03-08

**Overall Recommendation:** 3

**Summary:**

This paper presents ZipAR, a training-free framework for accelerating autoregressive visual generation. It leverages the local structure of images by allowing parallel decoding of spatially adjacent tokens, alongside the standard next-token prediction. An adaptive local window assignment with rejection sampling ensures contextual alignment. This method significantly reduces the number of forward passes needed for image generation—by up to 91% on the Emu3-Gen model—without requiring retraining, thus enhancing efficiency in visual generation tasks.

**Claims And Evidence:**

The claims made in the paper are clear and supported by experiments.

**Essential References Not Discussed:**

I think related work has been discussed in the paper.

**Experimental Designs Or Analyses:**

I think the experiments are a bit too few, because for the accelerated experiments, it would be better to add higher resolution tests to prove the effectiveness of the method in high resolution experiments. In addition, it would be good if the method can be effectively applied to other AR models.

**Methods And Evaluation Criteria:**

Perhaps we can add experiments with ImageNet 512 × 512, because generating images with higher resolutions often requires more acceleration. In addition, we can add more AR models to the evaluation, such as MAR.

**Other Comments Or Suggestions:**

No other suggestions.

**Other Strengths And Weaknesses:**

I think the advantage of this paper is that it provides motivation for the acceleration of AR model reasoning through methods such as attention map visualization analysis. And the acceleration module designed in this paper is reasonable and the effect is significant.

I think the disadvantage of this paper is mainly the lack of experiments with 512 resolution, which cannot illustrate the acceleration ability of the model at higher resolutions. In addition, the experiments in this paper are mainly implemented using the llama gen model, and lack verification of other models.

**Questions For Authors:**

Please provide a brief pseudo code to facilitate understanding of the sampling process.

**Relation To Broader Scientific Literature:**

The paper's use of rejection sampling and adaptive local window assignment draws from the concept of speculative decoding, which has been explored in natural language processing (NLP) to improve efficiency. Previous approaches, such as those used in language models, have shown that generating multiple tokens simultaneously can lead to significant efficiency gains. ZipAR applies this principle to visual generation, demonstrating that similar techniques can be effective beyond text.

**Theoretical Claims:**

There is nothing wrong with the theoretical statement.

---

> ### Author Rebuttal · Authors · 2025-03-31
>
> Thanks to the reviewer for the valuable comments.
>
> **Q1: The lack of experiments with 512 or higher resolution.**
> For clarity, we would like to highlight that our experimental results already include higher-resolution evaluations, as referred to Table 2 in the paper. Specifically, the LlamaGen-XL model operates at 512x512 resolution and the Lumina-mGPT model operates at 768x768 resolution. To improve transparency, we will explicitly state these resolutions in the revised Table 2 caption. Moreover, to further address your point, we have conducted additional experiments using the Lumina-mGPT-1024 model at 1024x1024 resolution on various benchmarks. Due to the character limit here, please refer to Q1 in our response to Reviewer ELf9 for the evaluation results. These results demonstrate our method's effectiveness across multiple resolution scales.
>
> **Q2: Experiments are mainly implemented using the LlamaGen model.**
> As referred to lines 299-300 in the paper, we integrate ZipAR with three state-of-the-art next-token AR visual generation models: LlamaGen, Lumina-mGPT and Emu3-Gen. Quantitative results can be found in Table 1-2 in the paper and the visualization results of these models can be found in Figures 1, 8 in the paper and Figures 9, 10 in the supplementary material.
>
> **Q3: Lack verification of other models, such as MAR.**
> It should be noted that ZipAR is a training-free, plug-and-play parallel decoding framework for **vanilla next-token AR** visual generation models. However, MAR does not follow a next-token prediction generation paradigm.
>
> **Q4: Provide a brief pseudo code for the sampling process.**
> Thanks for your valuable comment. We have provided a pseudo code for the sampling process, as shown below
>
> ```python
> # Pytorch-style Pseudo Code for ZipAR Sampling Process
>
> # Image latent dimensions: H x W
> # Minimum window size: s_min
>
> # Initialize variables
> total_columns = W                           # Number of columns
> total_rows = H                              # Number of rows
> decoding_rows = [0]                         # Rows actively being decoded
> decoded_tokens = [[] for _ in range(H)]     # Decoded tokens for each row
> pending_starts = []                         # Tentative new rows awaiting validation
>
> while decoding_rows or pending_starts:
>     # --- Step 1: Decode one token in each active row ---
>     for row in decoding_rows:
>         if len(decoded_tokens[row]) < total_columns:
>             # Decode next token using AR model
>             new_token = generate_token(row, len(decoded_tokens[row])) ## Generate token at position (row, len(decoded_tokens[row]))
>             decoded_tokens[row].append(new_token)
>
>     # --- Step 2: Process pending_rows
>     new_pending = []
>     for (new_row, old_token, old_prob) in pending_starts:
>         new_prob, new_token = speculative_generate(new_row, 0) ## Tentative generation of token at position (new_row, 0)
>         # Calculate acceptance probability
>         r = uniform_sample()
>
>         if r < min(1, new_prob[old_token] / old_prob[old_token]): ## Eq. 2 in the paper
>             # Accept token and start decoding
>             decoding_rows.append(new_row)
>             decoded_tokens[new_row].append(new_token)
>         else:
>             # Resample from difference distribution
>             resampled_prob, resampled_token = resample_distribution(new_prob, old_prob) ## Eq. 3 in the paper
>             new_pending.append( (new_row, resampled_token, resampled_prob) )
>
>     pending_starts = new_pending
>
>     # --- Step 3: Check for completed s_min-1 tokens to initiate new rows ---
>     for row in list(decoding_rows):
>         if len(decoded_tokens[row]) == s_min and \
>            row+1 < total_rows and \
>            row+1 not in decoding_rows and \
>            row+1 not in [p[0] for p in pending_starts]:
>             # Generate tentative token with window size s_min
>             prob, pend_token = speculative_generate(row+1, 0) ## Tentative generation of token at position (row+1, 0)
>             pending_starts.append( (row+1, pend_token, prob) )
>
>     # --- Step 4: Cleanup completed rows ---
>     decoding_rows = [r for r in decoding_rows
>                         if len(decoded_tokens[row]) < total_columns]
> ```

---

### Official Review · Reviewer_TPry · 2025-03-13

**Overall Recommendation:** 3

**Summary:**

This paper proposes ZipAR, a training-free method to accelerate the decoding speed of the AR image generation model. They first show that significant attention scores are allocated to tokens in the same column of previous rows. Therefore, decoding the next row is not necessary to wait for the finishing of the last row. Based on this idea, they design the ZipAR method and use an adaptive window size to control the number of tokens in one step. The results show that this method can accelerate the generation speed without a significant performance drop.

**Claims And Evidence:**

Yes

**Essential References Not Discussed:**

VAR published in NeurIPS 2024.

**Experimental Designs Or Analyses:**

1. Missing speed comparison with VAR and MaskGIT.
2. Need more benchmark to prove the robustness of ZipAR

**Methods And Evaluation Criteria:**

I think only evaluating text-to-image on MSCOCO with CLIP-Score is not robust enough. The quality results show no distinct difference between different models.

**Other Comments Or Suggestions:**

No

**Other Strengths And Weaknesses:**

Sec 3.3 is a little hard to understand intuitively the details.

**Questions For Authors:**

No

**Relation To Broader Scientific Literature:**

Accelerating current AR image generation

**Theoretical Claims:**

There is no proof of theoretical claims.

---

> ### Author Rebuttal · Authors · 2025-03-31
>
> Thanks to the reviewer for the valuable comments.
>
> **Q1: Essential reference VAR is not discussed.**
> As noted in our related work section (lines 157-160 in the paper), we do discuss VAR and its approach to visual generation. Moreover, it should be noted that VAR requires specialized multi-scale tokenizers and must be trained from scratch as a complete generation framework. In contrast, our proposed ZipAR is a training-free, plug-and-play parallel decoding solution for existing vanilla (raster order) next-token autoregressive visual generation models without any architectural modifications or retraining.
>
> **Q2: Missing speed comparison with VAR and MaskGIT.**
> As noted in Q1, ZipAR aims to accelerate existing vanilla next-token AR visual generation models, which is not directly comparable with VAR or MaskGIT. However, to address this concern, we have evaluated the generation efficiency of ZipAR, VAR and MaskGIT with similar model sizes. The results are presented below. Compared with vanilla next-token AR models, ZipAR greatly improves generation efficiency and narrows the efficiency gap with VAR and MaskGIT without any additional training.
>
> | Resolution | Model | Throughput (img/s) |
> | ---- | ---- | ---- |
> | 256x256 | LlamaGen-L | 40.9 |
> | 256x256 | ZipAR-11 | 47.0 |
> | 256x256 | ZipAR-7 | 58.1 |
> | 256x256 | ZipAR-3 | 80.8 |
> | 256x256 | MaskGIT | 120.0 |
> | 256x256 | VAR-d16 | 126.7 |
> | 512x512 | LlamaGen-L | 6.1 |
> | 512x512 | ZipAR-11 | 12.4 |
> | 512x512 | ZipAR-7 | 16.5 |
> | 512x512 | ZipAR-3 | 22.9 |
> | 512x512 | MaskGIT  | 50.8 |
> | 512x512 | VAR-d16  | 55.3 |
>
> **Q3: More benchmarks are needed to prove the robustness of ZipAR.**
> To address this concern, we have expanded our evaluation by assessing ZipAR’s performance using multiple metrics, including VQAScore, Human Preference Score v2, ImageReward, and Aesthetic Score, across three models: LlamaGen-XL-512, Lumina-mGPT-768, and Lumina-mGPT-1024. The results presented below demonstrate that our method significantly improves generation efficiency with little impact on output quality across various benchmarks.
>
> | Model | Method | Steps | VQAScore | HPSv2 | Image Reward | Aesthetic Score |
> | ---- | ---- | ---- | ---- | ---- | ---- | ---- |
> | LlamaGen-XL | NTP | 1024 | 0.6439 | **0.2647** | -0.0818 | 5.38 |
> | LlamaGen-XL | ZipAR-15 | 562 | 0.6534 | 0.2637 | **-0.0690** | **5.39** |
> | LlamaGen-XL | ZipAR-11 | 451 | **0.6581** | 0.2630 | -0.0982 | 5.37 |
> | LlamaGen-XL | ZipAR-7 | 324 | 0.6410 | 0.2625 | -0.1683 | 5.33 |
> | LlamaGen-XL | ZipAR-3 | 185 | 0.6343 | 0.2599 | -0.3121 | 5.32 |
> | Lumina-mGPT-768 | NTP | 2352 | 0.6579 | 0.2743 | **0.4164** | 6.10 |
> | Lumina-mGPT-768 | ZipAR-20 | 1063 | **0.6595** | **0.2747** | 0.3971 | **6.13** |
> | Lumina-mGPT-768 | ZipAR-17 | 915 | 0.6433 | 0.2732 | 0.3049 | 6.12 |
> | Lumina-mGPT-768 | ZipAR-14 | 740 | 0.6589 | 0.2739 | 0.3646 | 6.10 |
> | Lumina-mGPT-768 | ZipAR-11 | 588 | 0.6490 | 0.2730 | 0.2861 | 6.10 |
> | Lumina-mGPT-1024 | NTP | 4160 | 0.6718 | **0.2762** | **0.4232** | **5.97** |
> | Lumina-mGPT-1024 | ZipAR-20 | 1331 | 0.6705 | 0.2761 | 0.3913 | 5.95 |
> | Lumina-mGPT-1024 | ZipAR-17 | 1150 | **0.6797** | 0.2761 | 0.4018 | 5.94 |
> | Lumina-mGPT-1024 | ZipAR-14 | 964 | 0.6732 | 0.2747 | 0.3298 | 5.94 |
> | Lumina-mGPT-1024 | ZipAR-11 | 772 | 0.6723 | 0.2746 | 0.3222 | 5.95 |
>
> **Q4: Sec 3.3 can be more informative.**
> To clearly demonstrate the details of ZipAR, we have provided a pseudo code for ZipAR's sampling process. Please refer to Q4 in our response to Reviewer CC9X due to the character limit here. We will include it in the revised version.

---

### Official Review · Reviewer_ELf9 · 2025-03-13

**Overall Recommendation:** 3

**Summary:**

This paper introduces a novel technique to conduct parallel decoding in AR-based image generation. The proposed approach can be directly applied to off-the-shelf pretrained AR-based image generation models, speeding up the generation with small performance drop.

## update after rebuttal
Given the updated results with more evaluation metrics, I would like to keep my score of weak accept.

**Claims And Evidence:**

Yes, claims made in the submission are supported by clear and convincing evidence.

**Essential References Not Discussed:**

Related works are properly discussed.

**Experimental Designs Or Analyses:**

Yes, I have checked the soundness of all experimental designs and analyses. Overall, the experiments can validate the effectiveness of the proposed method. However, one issue is that this paper could benefit from more numerical results. Currently, only FID and CLIP-scores are provided.

The reviewer believe that some human evaluation results would enhance the significance of the paper. If human evaluation is not feasible, then at least more diverse automatic evaluation approach such as VQA-score [1], image reward [2] should be considered.

[1] Evaluating Text-to-Visual Generation with Image-to-Text Generation. Lin. et al.
[2] ImageReward: Learning and Evaluating Human Preferences for Text-to-Image Generation. Xu. et al.

**Methods And Evaluation Criteria:**

Yes, the proposed methods and evaluation makes sense.

**Other Comments Or Suggestions:**

Not applicable.

**Other Strengths And Weaknesses:**

Strength:
1) The proposed algorithm is simple and can be applied without the need of retraining.
2) The proposed approach is well-motivated and demonstrate promising results.

Weakness:
1) Please see "Experimental Designs Or Analyses".
2) The paper could also benefit from more ablation studies or discussions. For example, the author could study/discuss whether the proposed approach affects the optimal token-sampling-hyperparameters such as sampling temperature or CFG scale.

**Questions For Authors:**

Please see weakness (1), (2)

**Relation To Broader Scientific Literature:**

The key contributions can be related to the autoregressive-based image generation models. These models are known for their low generation speed. The proposed approach could alleviate such problem, and thus incentivize more researchers to explore AR-based image generation.

**Theoretical Claims:**

Not applicable.

---

> ### Author Rebuttal · Authors · 2025-03-31
>
> Thanks to the reviewer for the valuable comments.
>
> **Q1:More diverse automatic evaluation approach should be considered.**
> To address this concern, we have expanded our evaluation by assessing ZipAR’s performance using multiple metrics, including VQAScore, Human Preference Score v2, ImageReward, and Aesthetic Score, across three models: LlamaGen-XL-512, Lumina-mGPT-768, and Lumina-mGPT-1024. The results presented below demonstrate that our method significantly improves generation efficiency with little impact on output quality across various benchmarks.
> | Model | Method | Steps | VQAScore | HPSv2 | Image Reward | Aesthetic Score |
> | ---- | ---- | ---- | ---- | ---- | ---- | ---- |
> | LlamaGen-XL | NTP | 1024 | 0.6439 | **0.2647** | -0.0818 | 5.38 |
> | LlamaGen-XL | ZipAR-15 | 562 | 0.6534 | 0.2637 | **-0.0690** | **5.39** |
> | LlamaGen-XL | ZipAR-11 | 451 | **0.6581** | 0.2630 | -0.0982 | 5.37 |
> | LlamaGen-XL | ZipAR-7 | 324 | 0.6410 | 0.2625 | -0.1683 | 5.33 |
> | LlamaGen-XL | ZipAR-3 | 185 | 0.6343 | 0.2599 | -0.3121 | 5.32 |
> | Lumina-mGPT-768 | NTP | 2352 | 0.6579 | 0.2743 | **0.4164** | 6.10 |
> | Lumina-mGPT-768 | ZipAR-20 | 1063 | **0.6595** | **0.2747** | 0.3971 | **6.13** |
> | Lumina-mGPT-768 | ZipAR-17 | 915 | 0.6433 | 0.2732 | 0.3049 | 6.12 |
> | Lumina-mGPT-768 | ZipAR-14 | 740 | 0.6589 | 0.2739 | 0.3646 | 6.10 |
> | Lumina-mGPT-768 | ZipAR-11 | 588 | 0.6490 | 0.2730 | 0.2861 | 6.10 |
> | Lumina-mGPT-1024 | NTP | 4160 | 0.6718 | **0.2762** | **0.4232** | **5.97** |
> | Lumina-mGPT-1024 | ZipAR-20 | 1331 | 0.6705 | 0.2761 | 0.3913 | 5.95 |
> | Lumina-mGPT-1024 | ZipAR-17 | 1150 | **0.6797** | 0.2761 | 0.4018 | 5.94 |
> | Lumina-mGPT-1024 | ZipAR-14 | 964 | 0.6732 | 0.2747 | 0.3298 | 5.94 |
> | Lumina-mGPT-1024 | ZipAR-11 | 772 | 0.6723 | 0.2746 | 0.3222 | 5.95 |
>
> **Q2: Ablation studies on whether ZipAR affects the optimal token-sampling-hyperparameters.**
> We performed a grid search to determine the optimal token-sampling hyperparameters, namely, sampling temperature and classifier-free guidance scale, for ZipAR. The results are shown below. Here, "*" denotes the results obtained from LlamaGen's paper. These results indicate that ZipAR sampling does not alter the optimal sampling temperature and classifier-free guidance scale.
>
> | model | cfg | FID |
> | ---- | ---- | ---- |
> | LlamaGen-L* | 1.5 | 4.74 |
> | LlamaGen-L* | 1.75 | 3.15 |
> | LlamaGen-L* | 2.0 | **3.07** |
> | LlamaGen-L* | 2.25 | 3.62 |
> | ZipAR-16 | 1.5 | 6.18 |
> | ZipAR-16 | 1.75 | 3.72 |
> | ZipAR-16 | 2.0 | **3.14** |
> | ZipAR-16 | 2.25 | 3.44 |
>
> | model | Temperature | FID |
> | ---- | ---- | ---- |
> | LlamaGen-L | 0.96 | 3.53 |
> | LlamaGen-L | 0.98 | 3.24 |
> | LlamaGen-L* | 1.0 | **3.07** |
> | LlamaGen-L | 1.02 | 3.14 |
> | ZipAR-16 | 0.96 | 3.35 |
> | ZipAR-16 | 0.98 | 3.25 |
> | ZipAR-16 | 1.0 | **3.14** |
> | ZipAR-16 | 1.02 | 3.34 |

---

### Decision · Program_Chairs · 2025-05-01

**Decision:**

Accept (poster)

**Comment:**

The authors propose ZipAR, a training-free parallel decoding framework for accelerating autoregressive visual generation by leveraging parallel decoding of spatially adjacent tokens. The effectiveness of the proposed method is validated with three state-of-the-art autoregressive visual generation models (LlamaGen, Lumina-mGPT, and Emu3-Gen).

Initially, the paper received mixed scores. The reviewers were mainly concerned about the lack of more evaluation metrics, clarification of large resolution experiments, and speed comparisons with VAR and MaskGIT. The provided rebuttal effectively resolved the reviewers' concerns (with one minor unaddressed concern from Reviewer b59B regarding the artifacts in the generated results). As a result, the paper received all accept recommendations. After considering the author rebuttal and reviewer discussion/comments, the area chair agrees with this recommendation. The authors are encouraged to incorporate the rebuttal and reviewer suggestions to further improve the final version of the paper.